# Stimulation of Piezo1 by mechanical signals promotes bone anabolism

**Xuehua Li[1,2], Li Han[3], Intawat Nookaew[1,4], Erin Mannen[1,2], Matthew J Silva[5], Maria Almeida[1,2,3], Jinhu Xiong[1,2]***

[1]Center for Musculoskeletal Disease Research, University of Arkansas for Medical Sciences, Little Rock, United States; [2]Department of Orthopaedic Surgery, University of Arkansas for Medical Sciences, Little Rock, United States; [3]Division of Endocrinology, University of Arkansas for Medical Sciences, Little Rock, United States; [4]Department of Biomedical Informatics, University of Arkansas for Medical Sciences, Little Rock, United States; [5]Department of Orthopaedic Surgery, Washington University, St Louis, United States

**Abstract** Mechanical loading, such as caused by exercise, stimulates bone formation by osteoblasts and increases bone strength, but the mechanisms are poorly understood. Osteocytes reside in bone matrix, sense changes in mechanical load, and produce signals that alter bone formation by osteoblasts. We report that the ion channel Piezo1 is required for changes in gene expression induced by fluid shear stress in cultured osteocytes and stimulation of Piezo1 by a small molecule agonist is sufficient to replicate the effects of fluid flow on osteocytes. Conditional deletion of *Piezo1* in osteoblasts and osteocytes notably reduced bone mass and strength in mice. Conversely, administration of a Piezo1 agonist to adult mice increased bone mass, mimicking the effects of mechanical loading. These results demonstrate that Piezo1 is a mechanosensitive ion channel by which osteoblast lineage cells sense and respond to changes in mechanical load and identify a novel target for anabolic bone therapy.
DOI: https://doi.org/10.7554/eLife.49631.001

**\*For correspondence:**
JXIONG@uams.edu

## Introduction

Mechanical signals play critical roles in bone growth and homeostasis (*Turner et al., 2009*; *Ozcivici et al., 2010*). Mechanical stimuli increase bone mass by stimulating the activity and production of bone forming osteoblasts (*Meakin et al., 2014*; *Klein-Nulend et al., 2012*). In contrast, loss of mechanical signals decreases bone mass by reducing bone formation and stimulating production of bone resorbing osteoclasts (*Kondo et al., 2005*; *Nakamura et al., 2013*; *Xiong et al., 2011*). Osteocytes, which are cells buried in the bone matrix and derived from osteoblasts, are able to sense changes in mechanical load and orchestrate bone resorption and formation (*Bonewald, 2011*; *Klein-Nulend et al., 2013*). However, the molecular mechanisms by which osteocytes sense changes in mechanical loads remain unclear.

A variety of cell surface proteins and structures, including integrins, focal adhesions, and primary cilia, have been proposed to sense mechanical signals in bone cells (*Litzenberger et al., 2010*; *Nguyen and Jacobs, 2013*; *Rubin et al., 2006*). In addition, several lines of evidence suggest that ion channels are involved in the sensing of mechanical signals by osteocytes (*Hung et al., 1995*; *Lu et al., 2012*; *Lewis et al., 2017*; *Li et al., 2002*). For example, calcium influx is an early event following mechanical stimulus in osteocytes (*Hung et al., 1995*; *Lu et al., 2012*). Several calcium channels, including transient receptor potential channels (TRPV) and multimeric L-type and T-type voltage-sensitive calcium channels (VSCC) are expressed in osteoblasts and osteocytes (*Li et al., 2002*; *Abed et al., 2009*; *Shao et al., 2005*). TRPV4 is perhaps the most studied calcium channel in

**eLife digest** Bone size and strength depend on physical activity. Increased forces on the skeleton, such as those that occur during exercise, trigger more bone formation and make bones stronger. Conversely, reduced forces, caused for example by the lack physical activity, cause bone loss and increase the risk of fractures.

Bones contain cells called osteocytes. These cells sense the forces exerted on bone and orchestrate bone formation in response. Calcium channels are one type of molecule that has been proposed to help osteocytes to sense forces. Calcium channels reside in the cell membrane and can change their structure to allow calcium ions to flow into the cell. Some of them allow calcium ions into the cell in direct response to physical forces, communicating to the cell that a force has been applied. These are called mechanosensitive ion channels. Until now, however, no specific calcium channels involved in force sensing had been identified in osteocytes.

Li et al. searched for calcium channels in osteocytes, and found high levels of a mechanosensitive ion channel called Piezo1. Then, Li et al. made genetically modified mice that did not have any Piezo1 in these cells. The skeleton of these mice was small and weak. Moreover, the bones of these modified mice did not respond to forces like the bones of normal mice. To demonstrate this, Li et al. applied a short bout of increased force to the leg bones of unmodified mice and to those of the Piezo1 deficient mice. After two weeks, the bones of the unmodified mice had increased in thickness, whereas the bones lacking Piezo1 had not. A separate study by Sun, Chi et al. showed similar results when Piezo1 was removed from bone cells grown in the laboratory.

Finally, Li et al. tested the impact of a chemical called Yoda1 on bones. Yoda1 makes the Piezo1 channel open, thus mimicking a physical force. These experiments showed that mice treated with Yoda1 had thicker bones than untreated mice.

The ability of human bone to become stronger in response to exercise decreases with age, which contributes to the development of osteoporosis. Conditions that require severely restricted exercise, such as disability or extended bedrest, also lead to bone loss. These experiments show that Piezo1 allows bone to respond to physical force, and suggest Piezo1 as a promising therapeutic target to help curtail bone loss in these conditions.

DOI: https://doi.org/10.7554/eLife.49631.002

bone (*Lee et al., 2015*; *Masuyama et al., 2008*; *Mizoguchi et al., 2008*; *Suzuki et al., 2013*). Although conditional deletion of *Trpv4* in the osteoblast lineage has not yet been reported, *Trpv4* germline knockout mice exhibit high bone mass, which is opposite of what would be expected with loss of mechanical responsiveness (*Masuyama et al., 2008*; *van der Eerden et al., 2013*). Histological analysis of these mice revealed decreased osteoclast number and a normal bone formation rate (*Masuyama et al., 2008*; *van der Eerden et al., 2013*), arguing against a role for TRPV4 as a mechanosensor in bone. Although mice with germline deletion of the L-type VSCC *Cacna1d* have reduced cross-sectional area in long bones, these mice respond normally to mechanical loading (*Li et al., 2010*). Thus, heretofore, a definitive role for a specific calcium channel in the response of the skeleton to mechanical loading has not been demonstrated.

Herein we sought to identify calcium channels involved in mechanosensation in osteocytes. We found that *Piezo1*, a mechanosensitive ion channel, is highly expressed in osteocytes and its expression and activity were increased by fluid sheer stress. In addition, conditional deletion of *Piezo1* in osteoblasts and osteocytes decreased cortical thickness and cancellous bone volume. Moreover, the skeletal response to anabolic loading was significantly blunted in mice lacking *Piezo1* in osteoblasts and osteocytes. Importantly, administration of Yoda1, a Piezo1 agonist, increased bone mass in vivo. Overall, our results suggest that osteoblasts, osteocytes, or both, sense and respond to changes in mechanical signals in part via activation of the Piezo1 calcium channel and identify activation of Piezo1 signaling as a potential therapeutic approach for osteoporosis.

# Results

## Piezo1 mediates mechanotransduction in an osteocyte cell line

To identify calcium channels that respond to mechanical signals in osteocytes, we compared gene expression profiles of the osteocytic cell line MLO-Y4 under static and fluid flow conditions by RNA-seq. Principal components analysis and volcano plot of transcripts indicated that a significant number of genes were differentially expressed in MLO-Y4 cells under static versus fluid shear stress (*Figure 1—figure supplement 1A,B*). GO-enrichment analysis revealed enrichment in genes known to respond to mechanical signals, thereby validating the fluid flow experiment (*Figure 1—figure supplement 2A*). We then identified differentially expressed genes related to calcium channels. *Piezo1* was the most highly expressed among 78 calcium channels detected in MLO-Y4 cells under static condition (*Figure 1—figure supplement 2B*). In addition, *Piezo1* was also highly up-regulated by fluid flow in MLO-Y4 cells as determined by RNA-seq (*Figure 1A*) and RT-qPCR (*Figure 1B*). The Piezo ion channel family consists of two members, Piezo1 and Piezo2. While *Piezo2* is expressed predominately in neurons, *Piezo1* is mainly expressed in non-neuronal cells (*Murthy et al., 2017*). Consistent with this previous evidence, the expression of *Piezo1* was approximately 200-fold higher than

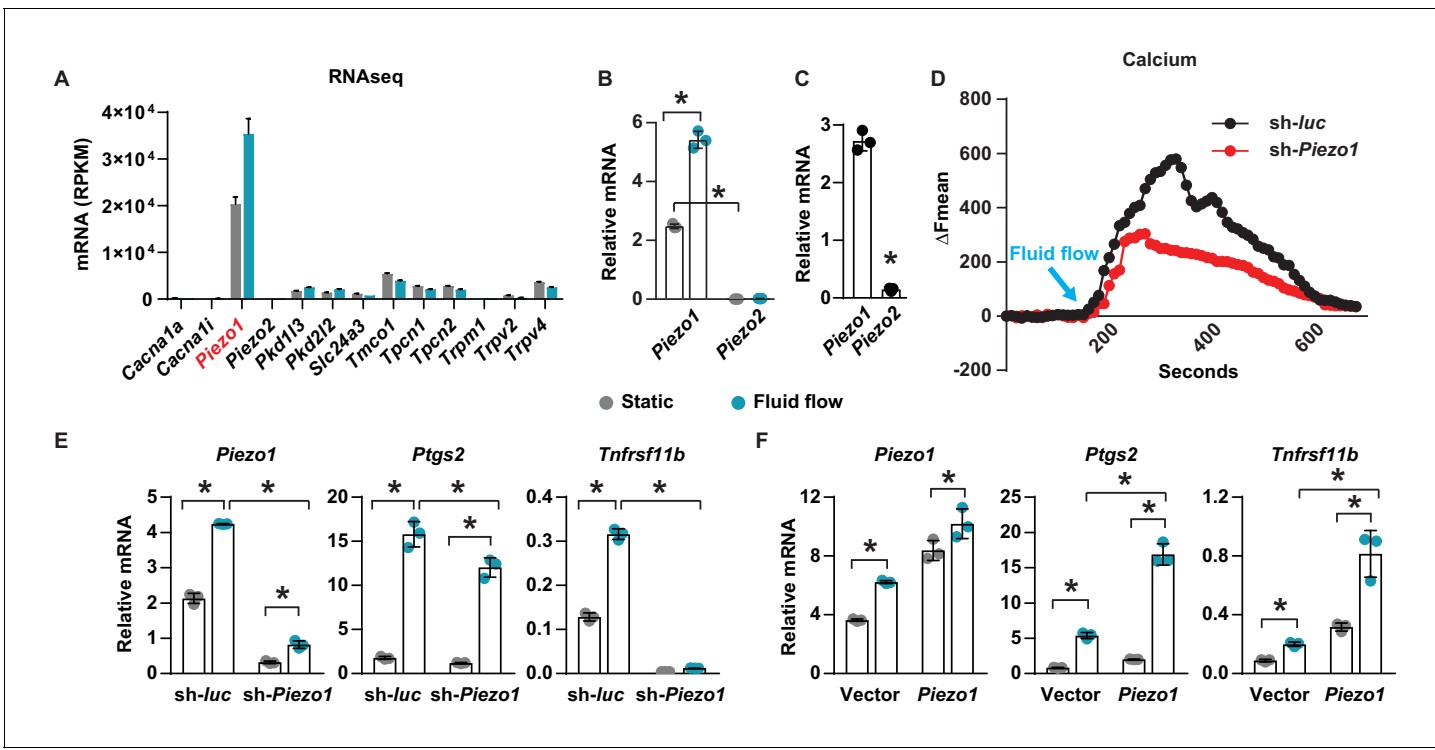

**Figure 1.** Piezo1 mediates mechanotransduction in an osteocyte cell line. (**A**) mRNA levels of calcium channels regulated by fluid shear stress in MLO-Y4 cells determined by RNA-seq (here and throughout, values are the mean ± s.d.). (**B**) qPCR of *Piezo1* and *Piezo2* mRNA in MLO-Y4 cells cultured under static or fluid shear stress conditions for 2 hr. *p<0.05 versus static, using Student's t-test. (**C**) *Piezo1* and *Piezo2* mRNA levels in cortical bone of 3-month-old wildtype C57BL/6J mice. (**D**) Intracellular calcium concentration measured in control or *Piezo1* knock-down MLO-Y4 cells before and after the start of fluid flow. Arrow indicates the time when fluid flow starts. (**E**) qPCR of *Piezo1*, *Ptgs2*, and *Tnfrsf11b* in control or *Piezo1* knock-down MLO-Y4 cells cultured under static or fluid shear stress conditions for 2 hr. n = 3 per group. (**F**) qPCR of *Piezo1*, *Ptgs2*, and *Tnfrsf11b* in control or *Piezo1* overexpressed MLO-Y4 cells cultured under static or fluid shear stress conditions for 2 hr. n = 3 per group. *p<0.05 with the comparisons indicated by the brackets using 2-way ANOVA. Gray indicates the static condition and teal indicates fluid shear stress.

DOI: https://doi.org/10.7554/eLife.49631.003

The following figure supplements are available for figure 1:

**Figure supplement 1.** Sequencing analysis of mRNA isolated from MLO-Y4 cells cultured under static or fluid shear stress conditions.
DOI: https://doi.org/10.7554/eLife.49631.004

**Figure supplement 2.** Sequencing analysis of mRNA isolated from MLO-Y4 cells cultured under static or fluid shear stress conditions.
DOI: https://doi.org/10.7554/eLife.49631.005

that of *Piezo2* in MLO-Y4 cells (*Figure 1B*). *Piezo1* expression was also much higher than *Piezo2* in osteocyte-enriched cortical bone isolated from 12-week-old mice (*Figure 1C*). Therefore, we focused our remaining analysis on *Piezo1*. Knock-down of *Piezo1* mRNA in MLO-Y4 cells significantly blunted the increase in intracellular calcium induced by fluid-flow (*Figure 1D*). Knock-down of *Piezo1* also blunted fluid-flow stimulation of *Ptgs2* and *Tnfrsf11b* (*Figure 1E*), two well-known targets of fluid shear stress in osteocytes (*Wadhwa et al., 2002*; *Zhao et al., 2016*). Conversely, overexpression of *Piezo1* in MLO-Y4 cells increased the expression of *Ptgs2* and *Tnfrsf11b* and enhanced their induction by fluid shear stress (*Figure 1F*). These results demonstrate that Piezo1 contributes to the response of MLO-Y4 cells to fluid shear stress.

## Loss of Piezo1 in osteoblasts and osteocytes decreases bone formation and bone mass

To determine the role of Piezo1 in osteocytes in vivo, we deleted *Piezo1* by crossing *Piezo1*^f/f mice (*Cahalan et al., 2015*) with *Dmp1-Cre* transgenic mice, which express the Cre recombinase in osteoblasts and osteocytes (*Bivi et al., 2012*; *Xiong et al., 2015*). Deletion of the *Piezo1* gene was confirmed by qPCR of genomic DNA isolated from osteocyte-enriched cortical bone (*Figure 2A*). Mice lacking the *Piezo1* gene in osteoblasts and osteocytes, hereafter referred to as *Dmp1-Cre;Piezo1*^f/f mice, exhibited normal body weight compared to their control *Piezo1*^f/f littermates (*Figure 2—figure supplement 1A*). Both female and male *Dmp1-Cre;Piezo1*^f/f mice exhibited low bone mineral density (BMD) at 5, 8, and 12 weeks of age as measured by dual energy x-ray absorptiometry (DXA) and the difference increased as the mice matured (*Figure 2B* and *Figure 2—figure supplement 1B*). Since the three control groups, including wild-type (WT), *Dmp1-Cre*, and *Piezo1*^f/f littermates, displayed similar BMD, we used *Piezo1*^f/f littermates as controls in the remaining studies. Spontaneous fractures were observed in the tibia of conditional knockout mice at a frequency of 0.16 (*Figure 2C*). Detailed analysis of the skeletal phenotype of these mice at 12 weeks of age by micro-CT revealed that femoral cortical thickness was lower in *Dmp1-Cre;Piezo1*^f/f mice compared with controls in both sexes (*Figure 2D,E* and *Figure 2—figure supplement 1C*). Periosteal and endocortical circumferences were also decreased in the femur of *Dmp1-Cre;Piezo1*^f/f mice (*Figure 2E*). In line with these changes, the total cross sectional area, cortical bone area, and medullary area were reduced in the conditional knockout mice (*Figure 2—figure supplement 1D*). In contrast to the changes in bone width, the length of the femurs was not different between genotypes indicating that longitudinal bone growth was normal in conditional knockout mice (*Figure 2—figure supplement 1E*). A decrease in cortical bone thickness was also detected in vertebrae of *Dmp1-Cre;Piezo1*^f/f female and male mice (*Figure 2F* and *Figure 2—figure supplement 1F*). Analysis of cancellous bone in the femur and vertebra revealed that bone volume over tissue volume, trabecular number, and trabecular thickness were decreased, while trabecular separation was increased in female *Dmp1-Cre;Piezo1*^f/f mice compared to their control littermates (*Figure 2G,H* and *Figure 2—figure supplement 1G,H*). Similar results were obtained in male mice (*Figure 2—figure supplement 1I,J*).

Biomechanical testing by 3-point bending showed that the femurs from *Dmp1-Cre;Piezo1*^f/f mice had reduced stiffness and ultimate force (*Figure 2I*). However, the Young's modulus and ultimate stress did not change, suggesting that the lower strength was due to differences in size and mass rather than changes in bone material properties (*Figure 2I*). Consistent with this, the tissue mineral density of femoral cortical bone was unaffected by deletion of *Piezo1* (*Figure 2J*).

To evaluate the cellular changes underlying the skeletal phenotype of the conditional knockout mice, we performed bone histomorphometry of femoral cortical bone and found that periosteal and endocortical mineralizing surfaces were significantly reduced in *Dmp1-Cre;Piezo1*^f/f mice at 5 weeks, an age of rapid bone growth (*Figure 2K* and *Figure 2—figure supplement 2A*). Bone formation at the outer (periosteal) surfaces of bone is a critical process for the enlargement of the skeleton. While double labels were easily seen in control mice, double labels were not observed in the conditional knockout mice, indicating that the bone formation rate at the periosteum of *Dmp1-Cre;Piezo1*^f/f mice was extremely low. Histomorphometric analysis of vertebral trabecular bone also revealed a decrease in mineralizing surface, mineral apposition rate, and bone formation rate in the conditional knockout mice (*Figure 2L*). In line with these changes, osteoblast number was lower in *Dmp1-Cre;Piezo1*^f/f mice (*Figure 2M*). In addition, we observed an increase in osteoclast number in the conditional knockout mice (*Figure 2M*).

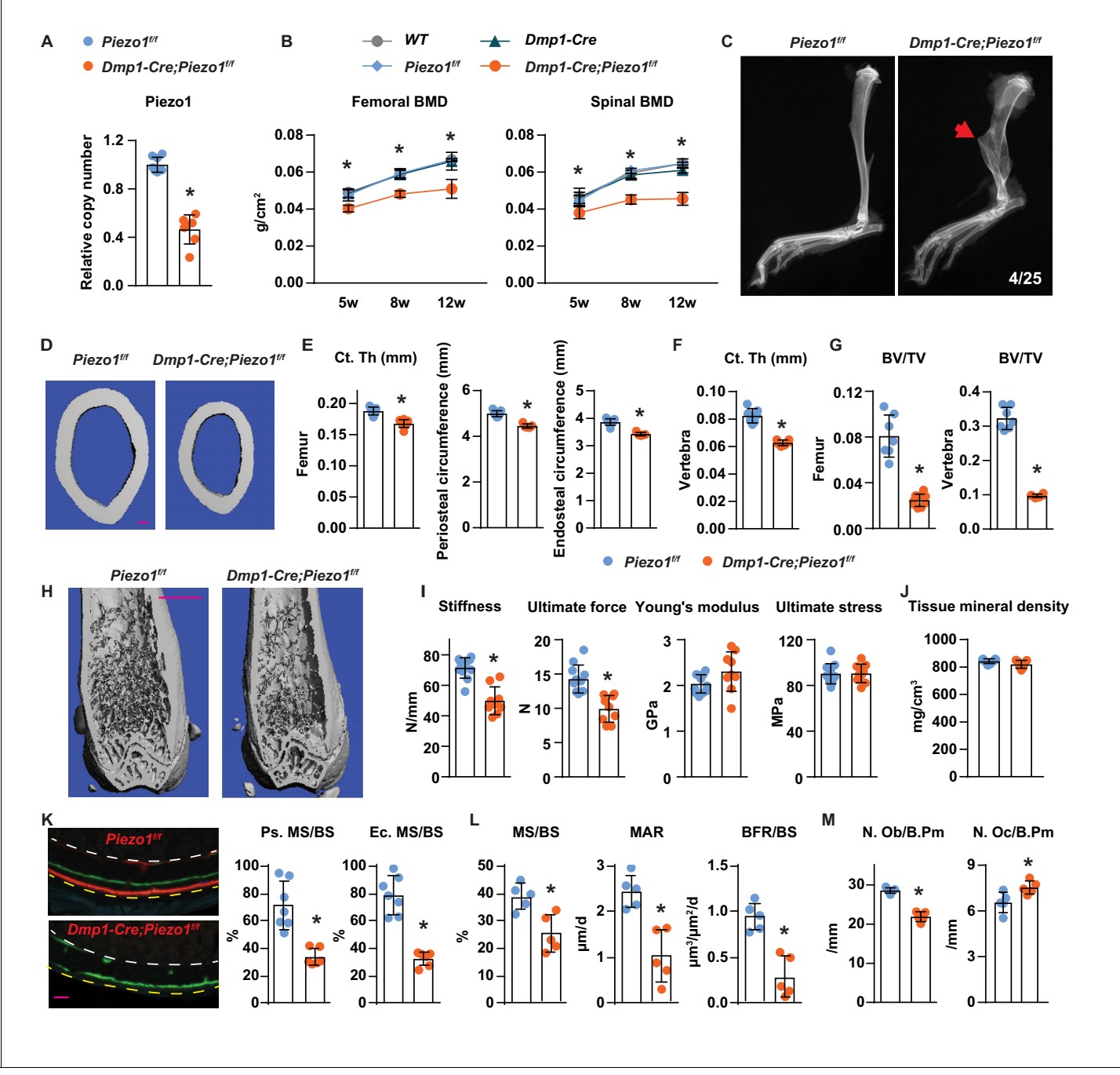

**Figure 2.** Loss of Piezo1 in osteoblasts and osteocytes decreases bone formation and bone mass. (A) qPCR of loxP-flanked *Piezo1* genomic DNA isolated from tibial cortical bone of *Dmp1-Cre;Piezo1*[f/f] (n = 6) and *Piezo1*[f/f] (n = 6) littermates. *p<0.05 using Student's t-test. (B) Serial BMD of female *Dmp1-Cre;Piezo1*[f/f] mice and their littermate controls at 5, 8, and 12 weeks of age. *p<0.05 using 2-way ANOVA at a given age. (C) X-ray images of tibia from 12-week-old *Dmp1-Cre;Piezo1*[f/f] and *Piezo1*[f/f] littermate. Arrowhead indicates the location of fracture. (D, E) Representative μCT images (scale bar, 0.1 mm) (D) and cortical thickness, periosteal circumference, and endocortical circumference analysis (E) of the femoral diaphysis in *Dmp1-Cre;Piezo1*[f/f] (n = 9) and *Piezo1*[f/f] (n = 9) littermates. (F) Cortical thickness measured in the 4th lumbar vertebra of 12-week-old female *Dmp1-Cre; Piezo1*[f/f] (n = 9) and *Piezo1*[f/f] (n = 9) littermates. (G) Bone volume per tissue volume (BV/TV) measured in the femur and the L4 vertebra of 12-week-old female *Dmp1-Cre;Piezo1*[f/f] (n = 9) and *Piezo1*[f/f] (n = 7) mice. (H) Representative μCT images of the distal femur. Scale bar, 1 mm. (I) Stiffness, ultimate force, Young's modulus, and ultimate stress measured in the femurs of *Dmp1-Cre;Piezo1*[f/f] (n = 9) and *Piezo1*[f/f] (n = 9) littermates. (J) Tissue mineral density measured in cortical bone in femoral diaphysis of *Dmp1-Cre;Piezo1*[f/f] (n = 9) and *Piezo1*[f/f] (n = 9) littermates. (K) Representative histological cross sections (left, yellow dotted line indicates periosteal surface and white dotted line indicates endocortical surface; scale bar = 100 μm) and quantification of mineralizing surface in periosteal and endocortical surface (right) at the femoral diaphysis of 5-week-old female *Dmp1-Cre;Piezo1*[f/f]

*Figure 2 continued on next page*

*Figure 2 continued*

($n$ = 7) and *Piezo1*$^{f/f}$ ($n$ = 5) littermates. (L, M) Mineralizing surface per bone surface (MS/BS), mineral apposition rate (MAR), and bone formation rate per bone surface (BFR/BS) (L), Osteoblast number (N.Ob/B.Pm), and osteoclast number (N.Oc/B.Pm) (M) measured in cancellous bone of lumbar vertebra 1–3 from 12-week-old female *Dmp1-Cre;Piezo1*$^{f/f}$ ($n$ = 5) and *Piezo1*$^{f/f}$ ($n$ = 5) littermates. *$p < 0.05$ using Student's t-test.

DOI: https://doi.org/10.7554/eLife.49631.006

The following figure supplements are available for figure 2:

**Figure supplement 1.** Loss of Piezo1 in osteoblasts and osteocytes decreases bone mass.

DOI: https://doi.org/10.7554/eLife.49631.007

**Figure supplement 2.** Deletion of *Piezo1* in osteoblasts and osteocytes decreases cortical bone.

DOI: https://doi.org/10.7554/eLife.49631.008

**Figure supplement 3.** Deletion of *Piezo1* from Dmp1-Cre-targeted cells does not affect muscle mass.

DOI: https://doi.org/10.7554/eLife.49631.009

To evaluate whether cell death could account for the changes seen with *Piezo1* deletion, we measured the percentage of empty osteocyte lacunae and osteocyte number. We did not observe changes in the percentage of empty osteocyte lacunae or the number of osteocytes normalized to bone area in *Dmp1-Cre;Piezo1*$^{f/f}$ mice compared to their littermate controls (*Figure 2—figure supplement 2B,C*). Consistent with these results, we did not observe any apparent morphological changes in osteocytes in the conditional knockout mice (*Figure 2—figure supplement 2D*). In addition, knock-down of *Piezo1* in MLO-Y4 cells decreased, rather than increased, Capase3 activity (*Figure 2—figure supplement 2E*). These results indicate that *Piezo1* deletion does not increase osteocyte death in vitro or in vivo. We also analyzed osteoblastogenesis in vitro and found normal osteoblast differentiation of bone marrow stromal cells from *Dmp1-Cre;Piezo1*$^{f/f}$ mice, as indicated by Alizarin Red staining (*Figure 2—figure supplement 2F*).

Since the *Dmp1-Cre* transgene also leads to recombination in a sub-population of muscle cells (*Lim et al., 2017*), we measured *Piezo1* deletion in gastrocnemius muscle, lean body weight, and gastrocnemius muscle mass to determine whether altered muscle mass could have contributed to the skeletal phenotype. We detected about 20% deletion of the *Piezo1* gene in the conditional knockout mice (*Figure 2—figure supplement 3A*). In addition, *Piezo1* expression in gastrocnemius muscle was about 10 times lower than that in bone (*Figure 2—figure supplement 3B*). More importantly, we did not observe any difference in lean body weight or gastrocnemius muscle mass between the conditional knockout mice and their control littermates (*Figure 2—figure supplement 3C,D*). These results demonstrate that Piezo1 in osteoblasts, osteocytes, or both, is essential for normal bone size and mass.

## Loss of Piezo1 in osteoblasts and osteocytes blunts the skeletal response to mechanical loads

To determine whether Piezo1 in osteoblasts or osteocytes is required for the skeletal response to increased mechanical loading, we loaded the left tibia of 16-week-old female *Dmp1-Cre;Piezo1*$^{f/f}$ mice and their control littermates with +1200με peak strain at the midshaft, as illustrated in *Figure 3A*. Two weeks of anabolic loading increased tibial cortical thickness in control mice but not in conditional knockout mice (*Figure 3B*). Consistent with the changes in bone mass, loading increased periosteal bone formation rate in control mice, due to increases in both mineralizing surface and mineral apposition rate (*Figure 3C,D*). The load-stimulated bone formation was significantly blunted in conditional knockout mice (*Figure 3C,D*). These results suggest that Piezo1 in osteoblasts, osteocytes, or both, plays an essential role in the response of the skeleton to mechanical loads.

## Piezo1 controls *Wnt1* expression via YAP1 and TAZ

To understand the molecular mechanisms by which Piezo1 increases bone mass, we compared expression of genes known to influence bone formation and resorption between *Dmp1-Cre;Piezo1*$^{f/f}$ mice and control littermates. Production of Wnt1 or the Wnt signaling inhibitor Sclerostin (*Sost*) by osteocytes represent critical stimulatory or inhibitory signals to bone formation, respectively (*Luther et al., 2018*; *Li et al., 2008*). *Wnt1* mRNA was lower in cortical bone shafts of conditional knockout mice at both 5 and 12 weeks of age while the expression of *Sost* was unaffected

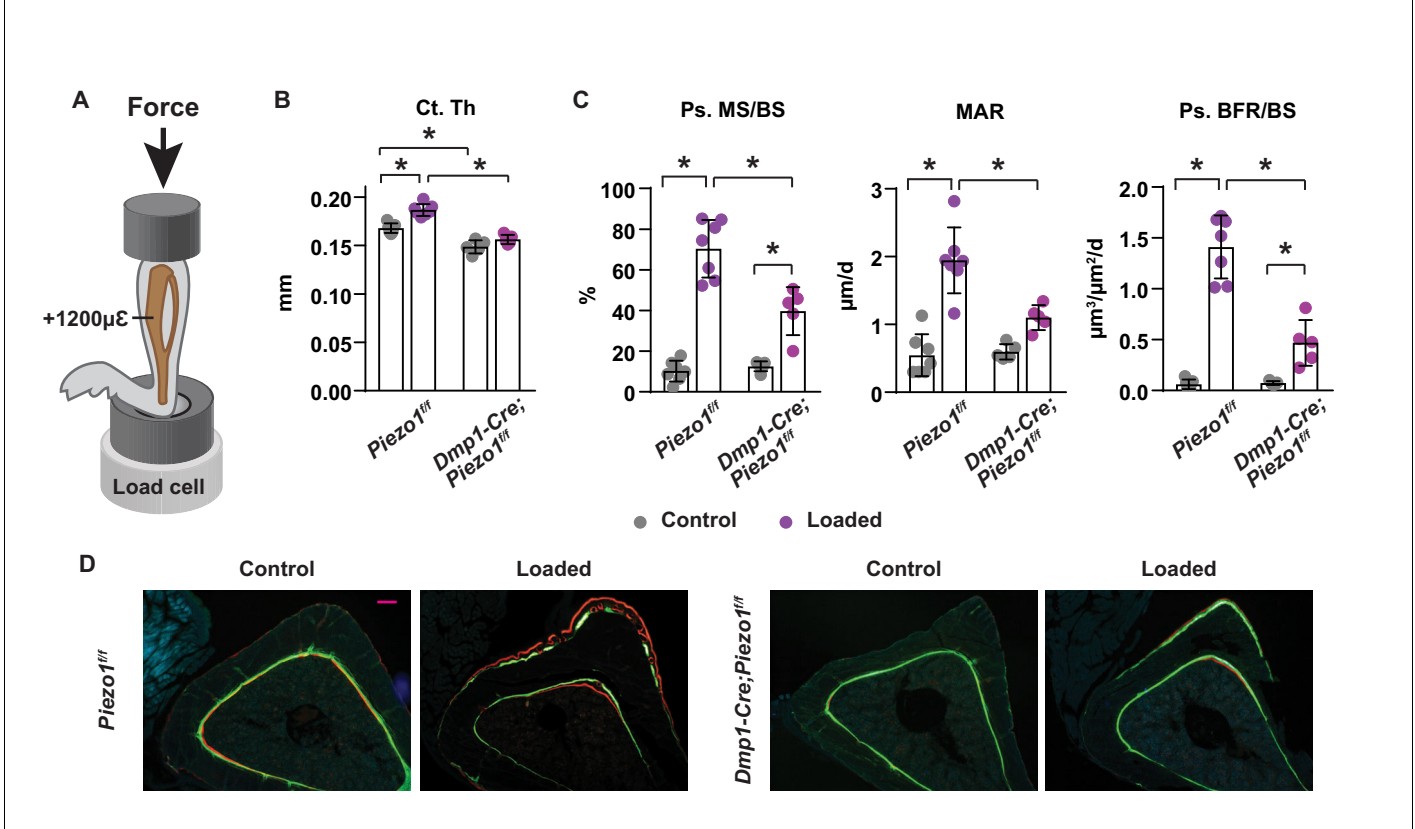

**Figure 3.** Loss of Piezo1 in osteoblasts and osteocytes blunts the skeletal response to mechanical loads. (A) Schematic illustration of anabolic loading on mouse tibia. (B) Cortical thickness (Ct.Th) in the tibial shaft of 4-month-old loaded or control *Dmp1-Cre;Piezo1^{f/f}* (n = 5) and *Piezo1^{f/f}* (n = 7) littermates. (C) Mineralizing surface (MS/BS), mineral apposition rate (MAR), and bone formation rate (BFR/BS) in periosteal surface of the tibia of 4-month-old female *Dmp1-Cre;Piezo1^{f/f}* (n = 5) and *Piezo1^{f/f}* (n = 7) littermates. (D) Representative histological cross section images of the tibial shaft of 4-month-old female *Dmp1-Cre;Piezo1^{f/f}* and *Piezo1^{f/f}* littermates. Scale bar, 100 μm. *p<0.05 with the comparisons indicated by the brackets using 2-way ANOVA.

DOI: https://doi.org/10.7554/eLife.49631.010

(*Figure 4A,B*). Consistent with increased osteoclast number, expression of the essential pro-osteoclastogenic cytokine RANKL (*Tnfsf11*) was higher in the conditional knockout mice (*Figure 4B*). In contrast, expression of OPG (*Tnfrsf11b*), a secreted decoy receptor for RANKL, was not different between the genotypes (*Figure 4B*), despite our observation of reduced *Tnfrsf11b* expression in MLO-Y4 cells lacking *Piezo1* (*Figure 1E*).

The expression of *Wnt1* can be stimulated by mechanical loading in mice (*Holguin et al., 2016*). Therefore, we determined whether mechanical signals increase *Wnt1* expression via *Piezo1*. Fluid shear stress increased *Wnt1* expression in MLO-Y4 cells but this was blunted after knock-down of *Piezo1* (*Figure 4C*). Basal expression of *Wnt1* was also reduced by *Piezo1* knock-down (*Figure 4C*). YAP1 and TAZ are two related transcriptional cofactors that can be activated by mechanical signals, including fluid flow and matrix rigidity, and recently Piezo1 has been shown to control their activity (*Wang et al., 2016*; *Dupont et al., 2011*; *Pathak et al., 2014*). We have shown previously that deletion of *Yap1* and *Taz* using *Dmp1-Cre* decreases bone mass, due to both reduced bone formation and increased osteoclast number (*Xiong et al., 2018*). Here, we analyzed the diaphysis of femurs of these mice and found that cortical thickness, periosteal circumference, and endocortical circumference were significantly decreased in *Dmp1-Cre;Yap1^{f/f},Taz^{f/f}* mice compared to their *Yap1^{f/f},Taz^{f/f}* littermates (*Figure 4—figure supplement 1A*). Because these changes were similar to the ones seen in cortical bone of *Dmp1-Cre;Piezo1^{f/f}* mice, we examined whether Piezo1 controls *Wnt1* expression via YAP1 and TAZ. Silencing the *Piezo1* gene in MLO-Y4 cells decreased the expression of *Cyr61*, a YAP1 and TAZ target gene, and blunted fluid shear stress induction of *Cyr61* expression

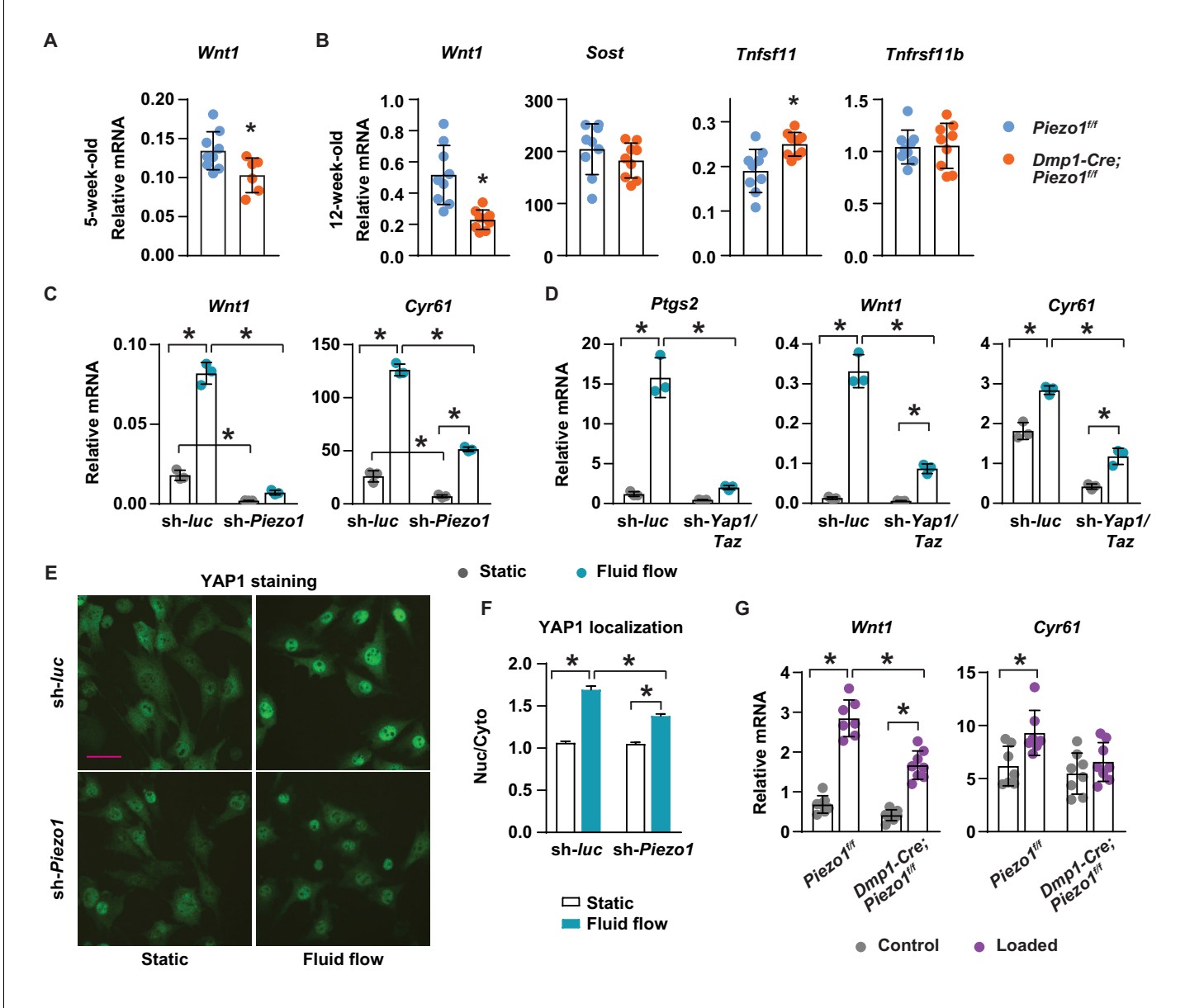

**Figure 4.** Piezo1 controls *Wnt1* expression via YAP1 and TAZ. (**A**) qPCR of *Wnt1* mRNA in tibial cortical bone of 5-week-old female *Piezo1^f/f* (n = 6) and *Dmp1-Cre;Piezo1^f/f* mice (n = 6). *p<0.05 using Student's t-test. (**B**) Relative mRNA levels of *Wnt1, Sost, Tnfsf11 (RANKL),* and *Tnfrsf11b (OPG)* in tibia cortical bone of 12-week-old female *Piezo1^f/f* (n = 9) and *Dmp1-Cre;Piezo1^f/f* (n = 9) mice. *p<0.05 using Student's t-test. (**C**) *Wnt1* and *Cyr61* mRNA levels in control or *Piezo1* knock-down MLO-Y4 cells cultured under static or fluid shear stress conditions. *p<0.05 with the comparisons indicated by the brackets using 2-way ANOVA. (**D**) *Ptgs2, Wnt1,* and *Cyr61* mRNA levels in control or *Yap1/Taz* knock-down MLO-Y4 cells cultured under static or fluid shear stress conditions. *p<0.05 with the comparisons indicated by the brackets using 2-way ANOVA. (**E**) YAP1 immunofluorescence in control or *Piezo1* knock-down MLO-Y4 cells cultured under static or fluid shear stress conditions. Scale bar, 100 μm. (**F**) Quantification of mean fluorescence intensity in nucleus versus cytoplasm in the cells described in (**E**). (**G**) *Wnt1* and *Cyr61* mRNA levels measured in tibia of female *Dmp1-Cre;Piezo1^f/f* (n = 8) and *Piezo1^f/f* (n = 7) mice loaded with one bout of compressive loading. Mice were harvested 5 hr after loading. *p<0.05 with the comparisons indicated by the brackets using 2-way ANOVA.

DOI: https://doi.org/10.7554/eLife.49631.011

The following figure supplements are available for figure 4:

**Figure supplement 1.** Loss of YAP1 and TAZ in osteoblasts and osteocytes decreases cortical bone.
DOI: https://doi.org/10.7554/eLife.49631.012
**Figure supplement 2.** Deletion of *Piezo1* in osteoblastic cells blunts their response to fluid flow.
DOI: https://doi.org/10.7554/eLife.49631.013

(*Figure 4C*). We then silenced the *Yap1* and *Taz* genes in MLO-Y4 cells to examine whether these factors are required for the stimulation of *Wnt1* by fluid shear stress. We found that lack of *Yap1* and *Taz* blunted the response to fluid flow including the increase in *Ptgs2*, *Wnt1*, and *Cyr61* expression (*Figure 4D*). Knock-down of *Piezo1* and *Yap1/Taz* was confirmed by mRNA abundance (*Figure 4—figure supplement 1B,C*). Importantly, silencing *Piezo1* blunted YAP1 activation caused by fluid shear stress, indicated by blunted nuclear translocation of YAP1 (*Figure 4E,F*). Similarly, we deleted *Piezo1* in UAMS-32 cells, a murine osteoblastic cell line, using CRISPR/Cas9 and found that expression of *Ptgs2*, *Wnt1*, and *Cyr61* induced by fluid flow were blunted in *Piezo1* knock out cells (*Figure 4—figure supplement 2*). To determine whether Piezo1 is required for *Wnt1* expression induced by mechanical loading in vivo, we applied one bout of compressive loading on the tibia of *Dmp1-Cre;Piezo1*$^{f/f}$ mice and their *Piezo1*$^{f/f}$ littermates with +1200με peak strain at the midshaft. Mechanical loading increased *Wnt1* and *Cyr61* expression in control mice (*Figure 4G*). However, these increases were blunted in *Dmp1-Cre;Piezo1*$^{f/f}$ mice (*Figure 4G*). Taken together, these results indicate that stimulation of Piezo1 by mechanical signals increases *Wnt1* expression at least in part via activation of YAP1 and TAZ.

## Activation of Piezo1 mimics the effects of mechanical stimulation on osteocytes

Finally, we determined whether activation of Piezo1 is sufficient to mimic the effects of mechanical stimulation in osteocytes and bone. Treatment of MLO-Y4 cells with Yoda1, a small molecule agonist of Piezo1 (*Syeda et al., 2015*), increased intracellular calcium concentration (*Figure 5A*), and stimulated expression of *Ptgs2*, *Wnt1*, and *Tnfrsf11b* (*Figure 5B*), similar to the effect of fluid flow on these cells. Importantly, silencing of *Piezo1* completely prevented the increase of intracellular calcium (*Figure 5A*), as well as the changes in gene expression induced by Yoda1 (*Figure 5B*). Likewise, silencing *Yap1* and *Taz* in MLO-Y4 cells significantly blunted the increase of *Ptgs2*, *Wnt1*, and *Tnfrsf11b* by Yoda1, indicating that the response to Yoda1 also requires YAP1 and TAZ (*Figure 5C*). Yoda1 also promoted expression of *Ptgs2*, *Wnt1*, *Tnfrsf11b*, *Cyr61*, and decreased *Sost* in cortical bone organ cultures from C57BL/6J mice (*Figure 5D*). Importantly, Yoda1 increased *Wnt1* expression in osteocyte-enriched cortical bone in vivo (*Figure 5E*). These results demonstrated that Yoda1 mimics the response to fluid flow in authentic osteocytes.

To determine whether Yoda1 is able to increase bone mass in vivo, we administered Yoda1 to 4-month-old female WT C57BL/6J mice for 2 weeks (*Figure 5F*). Yoda1 did not alter body weight (*Figure 5—figure supplement 1A*) but increased cortical thickness and cancellous bone mass in the distal femur (*Figure 5G*). Yoda1 also increased cortical thickness in the vertebra (*Figure 5H*). However, we did not detect changes in cancellous bone volume in vertebrae (*Figure 5H*). Consistent with the effect on bone mass, the serum levels of osteocalcin, a bone formation marker, were increased in Yoda1-treated mice (*Figure 5I*). In contrast, we did not observe changes in the serum levels of CTX, a bone resorption marker (*Figure 5—figure supplement 1B*). Our results demonstrate that activation of Piezo1 by Yoda1 mimics the effects of fluid shear stress on osteocytes and increases bone mass in mice.

## Discussion

Loss of function studies in epithelial cells have shown that Piezo1 responds to various forms of mechanical forces, including membrane stretch, static pressure, and fluid shear stress (*Li et al., 2014*; *Gudipaty et al., 2017*; *Miyamoto et al., 2014*). Moreover, Piezo1 can be activated by mechanical perturbations of the lipid bilayer alone, demonstrating its role in mechanosensation (*Syeda et al., 2016*). Here, the rapid response of MLO-Y4 cells to fluid shear stress is blunted by knocking-down Piezo1 indicating its important role in mechanosensation in bone cells. In addition, the basal skeletal phenotype of mice lacking *Piezo1* in osteoblasts and osteocytes suggests that they have a reduced ability to respond to mechanical stimulation. Direct testing of this idea by performing an anabolic loading regime confirmed that the bones of the conditional knockout mice were less responsive to mechanical signals than controls. This decrease cannot be attributed to intrinsic cell defect since cell survive is not affected by Piezo1 deletion. Thus, our studies demonstrate that Piezo1 plays a critical role in sensing mechanical signals and maintaining bone homeostasis. In humans, truncation mutations in *Piezo1* cause a recessive form of generalized lymphatic dysplasia

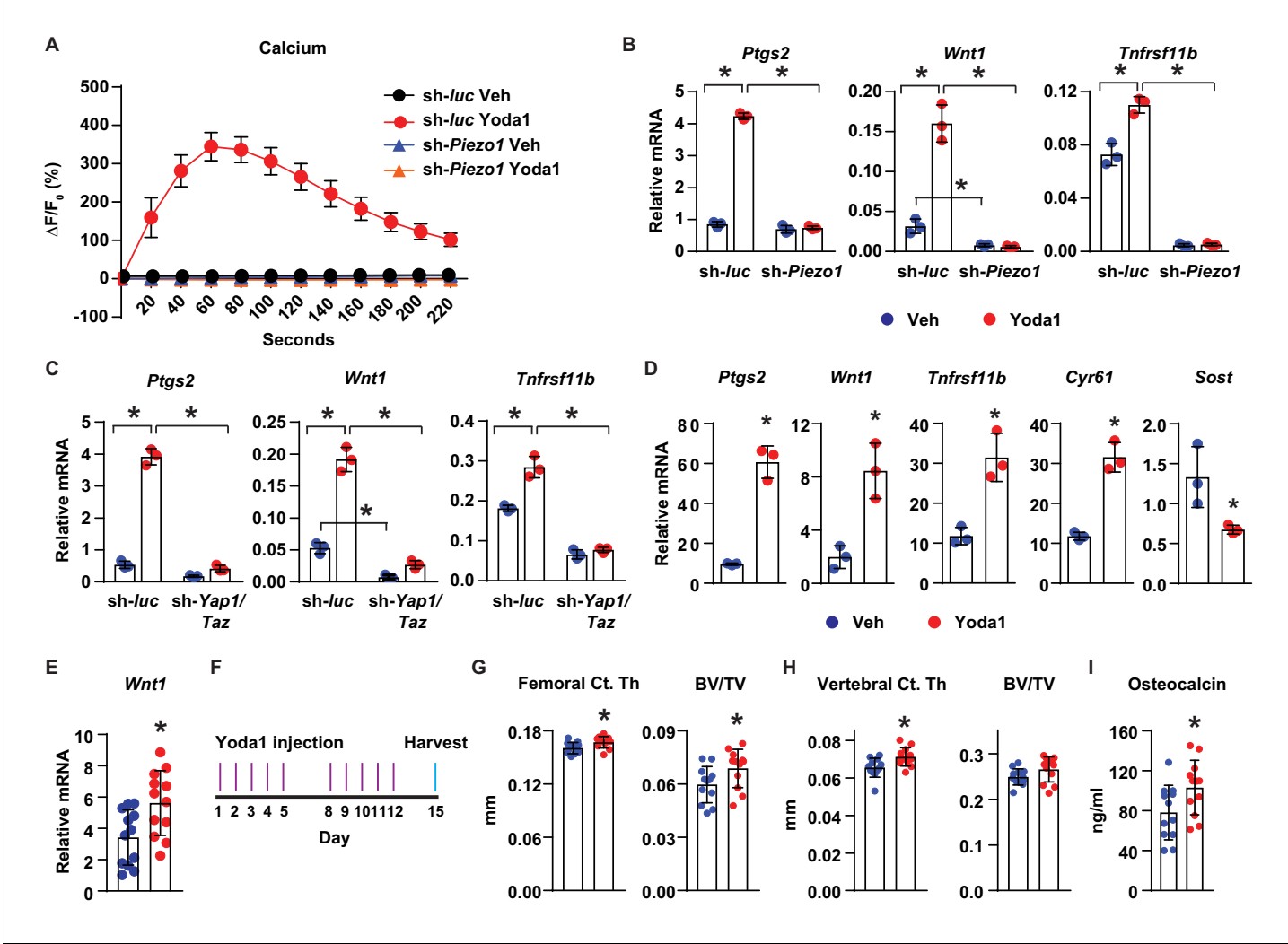

**Figure 5.** Activation of Piezo1 mimics the effects of mechanical stimulation on osteocytes. (**A**) Intracellular calcium concentration measured in control or *Piezo1* knock-down MLO-Y4 cells immediately after the treatment of DMSO or 10 μM Yoda1. (**B**) qPCR of *Ptgs2*, *Wnt1*, and *Tnfrsf11b* in control or *Piezo1* knock-down MLO-Y4 cells treated with DMSO or 10 μM Yoda1 for 2 hr. n = 3 per group. *p<0.05 versus vehicle treated controls of the same genotype by 2-way ANOVA. (**C**) qPCR of *Ptgs2*, *Wnt1*, and *Tnfrsf11b* in control or *Yap1/Taz* knock-down MLO-Y4 cells treated with DMSO or 10 μM Yoda1 for 2 hr. n = 3 per group. *p<0.05 versus vehicle treated controls of the same genotype by 2-way ANOVA. (**D**) qPCR of *Ptgs2*, *Wnt1*, *Tnfrsf11b*, *Cyr61*, and *Sost* in ex vivo cultured femoral cortical bone from 5-week-old mice treated with DMSO or 10 μM Yoda1 for 4 hr. n = 3 per group. (**E**) qPCR of *Wnt1* in tibia of C57BL/6J mice treated with Veh or Yoda1 for 4 hr. n = 12 per group. (**F**) Schedule of in vivo Yoda1 administration. (**G, H**) Cortical thickness and cancellous BV/TV in distal femur (**G**) and the 4th lumbar (**H**) of 4-month-old vehicle or Yoda1 treated female C57BL/6J mice (n = 12 per group). (**I**) Circulating osteocalcin levels in the serum of 4-month-old vehicle or Yoda1 treated female C57BL/6J mice (n = 12 per group). *p<0.05 versus vehicle treated controls by Student's t-test.

DOI: https://doi.org/10.7554/eLife.49631.014

The following figure supplement is available for figure 5:

**Figure supplement 1.** Yoda1 does not affect body weight and serum bone resorption marker.
DOI: https://doi.org/10.7554/eLife.49631.015

but a musculoskeletal phenotype has not been reported (*Fotiou et al., 2015*). Nonetheless, SNPs in the human *Piezo1* locus are associated with low bone mineral density and increased fracture risk (*Morris et al., 2019*).

While preparing the revision of this manuscript, Sun et, al published a similar study in which *Piezo1* was deleted from osteoblast lineage cells using *BGLAP-Cre* transgenic mice (*Sun et al., 2019*). Similar to our studies, deletion of *Piezo1* in osteoblast lineage cells resulted in a low bone

mass phenotype. Importantly, loss of Piezo1 in osteoblast lineage cells blunted the bone loss caused by hind-limb suspension, supporting the idea that Piezo1 contributes to the skeletal response to mechanical stimulation.

Deletion of Piezo1 from osteoblasts and osteocytes did not completely abolish the response of skeleton to mechanical stimulus. Thus Piezo1 is not the sole mechanosensor in osteoblasts and osteocytes. Other cell surface proteins and structures including integrins, focal adhesions, and primary cilia, also likely contribute to sensing mechanical signals in bone. Possible crosstalk between Piezo1 and these other sensors will need to be addressed in future studies. It is also possible that cells other than osteoblasts and osteocytes, such as osteoblast progenitors, sense changes in load and contribute to the increase in bone formation.

It is important to note that, in addition to osteoblasts and osteocytes, the *Dmp1-Cre* transgene used in our study also causes recombination in skeletal muscle cells (*Xiong et al., 2011*; *Lim et al., 2017*; *Xiong et al., 2015*). Therefore, it is possible that loss of *Piezo1* in muscle cells also contributed to the skeletal phenotype we observed in the conditional knockout mice. However, lean body weight and muscle mass in the conditional knockout mice were unchanged, arguing against a role for muscle cells in the skeletal phenotype. In addition, the potent effects of *Piezo1* gain- and loss-of-function in MLO-Y4 cells suggest that its effects are at least partly due to actions in osteocytes. Nonetheless, to distinguish between the possible contributions of Piezo1 in osteoblasts versus osteocytes, further studies using a Cre driver strain that is active in osteocytes but not in osteoblasts will be required.

We identified Wnt1 as a potential downstream effector of Piezo1. Previous studies have shown that mechanical loading increases *Wnt1* expression in murine bone (*Holguin et al., 2016*; *Kelly et al., 2016*). Importantly, deletion of *Wnt1* in osteoblasts and osteocytes using a *Dmp1-Cre* transgene produced a skeletal phenotype that resembles the one we observed by deletion of *Piezo1* using the same Cre driver strain (*Joeng et al., 2017*). Taken together, these results suggest that mechanical signals stimulate *Wnt1* expression via activation of Piezo1. The molecular pathways by which Piezo1 controls gene expression are only partially understood. Nonetheless, cell culture studies demonstrate that Piezo1 is required for YAP1 nuclear localization in neural stem cells (*Pathak et al., 2014*). Consistent with this, we found that Piezo1 controls nuclear translocation of YAP1 induced by fluid flow in MLO-Y4 cells. YAP1 and TAZ have been implicated as mediators of the response to mechanical signals in a variety of cell types (*Dupont et al., 2011*; *Hansen et al., 2015*). Our finding that YAP1 and TAZ are required for stimulation of *Wnt1* by fluid flow or Yoda1 suggests that mechanical activation of Piezo1 stimulates *Wnt1* expression in osteocytes, at least in part, by activating YAP1 and TAZ. Consistent with this idea, deletion of *Yap1* and *Taz* in mature osteoblasts and osteocytes caused a skeletal phenotype that was similar to deletion of *Piezo1*, albeit less pronounced (*Xiong et al., 2018*). The milder bone phenotype of *Yap1/Taz* conditional knockout mice suggests that YAP1 and TAZ are not the only downstream effectors of Piezo1 in osteoblast lineage cells.

Similar to unloading, deletion of *Piezo1* in osteoblasts and osteocytes led to not only decreases in bone formation, but also increases in RANKL expression and bone resorption. Indeed, increased RANKL expression as well as osteoclast number have been observed in hind-limb unloaded mice (*Xiong et al., 2011*). In our previous studies, we detected an increase in osteoclast number in mice that lack *Yap1* and *Taz* in osteoblasts and osteocytes (*Xiong et al., 2018*), suggesting that YAP1 and TAZ are downstream effectors of Piezo1 in controlling osteoclast formation. Thus, loss of Piezo1 in osteoblasts and osteocytes mimics the overall effect of unloading on the skeleton, further supporting the idea that Piezo1 is a mechanosensor in bone.

Activation of Piezo1 using the small molecule Yoda1 mimics the effects of fluid flow in various cell types including endothelial cells, erythrocytes, platelets, and smooth muscle cells (*Cahalan et al., 2015*; *Li et al., 2014*; *Ilkan et al., 2017*; *Rode et al., 2017*). In addition, Yoda1 administration promotes lymphatic valve formation during development (*Choi et al., 2019*). Here, we showed that Piezo1 activation by Yoda1 mimics the impact of mechanical stimulation in cultured osteocytic cells as well as ex vivo bone organ cultures. More importantly, administration of Yoda1 to mice increased bone mass and elevated a bone formation marker in the circulation, demonstrating that activation of Piezo1 is a potential target for anabolic bone therapy. One possible limitation of such an approach would be the functions of Piezo1 in other tissues, such as the vasculature. However, it is important to note that bone anabolism requires only transient mechanical stimulation of the skeleton in rodents

or humans (*Vlachopoulos et al., 2018*; *Hinton et al., 2015*; *Kato et al., 2006*). Therefore, it is possible that selectivity for bone anabolism may be achieved by administration regimes that result in only transient activation of Piezo1 by ligands such as Yoda1.

In summary, our studies demonstrate a critical role for Piezo1 in the maintenance of bone homeostasis and suggest that this occurs via mediation of mechanosensation in osteoblasts, osteocytes, or both. Our finding that activation of Piezo1 mimics the effects of mechanical stimulation on bone cells and increases bone mass in mice sets the stage for exploration of this pathway as a therapeutic target for osteoporosis.

# Materials and methods

## Key resources table

| Reagent type (species) or resource | Designation | Source or reference | Identifiers | Additional information |
|---|---|---|---|---|
| Genetic reagent (*M. musculus*) | Mouse: Piezo1$^{f/f}$ (*Piezo1$^{tm2.1Apat}$*/J) | Jackson Laboratories | JAX: 029213; RRID:IMSR_JAX:029213 | |
| Genetic reagent (*M. musculus*) | Mouse: *Dmp1-Cre* | *Bivi et al., 2012* | N/A | |
| Genetic reagent (*M. musculus*) | Mouse: *Yap1$^{f/f}$;Taz$^{f/f}$* | *Xin et al., 2013* | N/A | |
| Genetic reagent (*M. musculus*) | Mouse: WT C57BL/6J | Jackson Laboratories | JAX: 000664; RRID:IMSR_JAX:000664 | |
| Commercial assay or kit | Mouse Osteocalcin Immunoassay Kit | Thermo Fisher | Cat# J64239 | |
| Commercial assay or kit | Fluo-8 Calcium Flux Assay Kit | Abcam | Cat# ab112129 | |
| Commercial assay or kit | RatLaps (CTX-I) EIA kit | Immunodiagnostic Systems | Cat# AC-06F1 | |
| Commercial assay or kit | TruSeq stranded mRNA kit | Illumina | Cat# 20020594 | |
| Commercial assay or kit | High-capacity cDNA reverse transcription kit | Life Technologies | Cat# 4368813 | |
| Commercial assay or kit | RNeasy mini kit | QIAGEN | Cat# 74106 | |
| Cell line (Murine) | 293T | ATCC | CRL-3216 | |
| Cell line (Murine) | MLO-Y4 | *Kato et al., 1997* | | |
| Cell line (Murine) | UAMS-32 | *O'Brien et al., 1999* | | Cell line maintained in Charles O'Brien lab |
| Transfected construct (*M. musculus*) | *Piezo1* shRNA forward | *Zhang et al., 2017* | Oligo | CCGGTCGGCGCTTGCTAGAA CTTCACTCGAGTGAAGTTC TAGCAAGCGCCGATTTTTG |
| Transfected construct (*M. musculus*) | *Piezo1* shRNA reverse | *Zhang et al., 2017* | Oligo | AATTCAAAAATCGGCGCTTG CTAGAACTTCACTCGAGTGAA GTTCTAGCAAGCGCCGA |
| Transfected construct (*M. musculus*) | *Yap1* shRNA | Sigma-Aldrich | TRCN0000238432 | |
| Transfected construct (*M. musculus*) | *Taz* shRNA | Sigma-Aldrich | TRCN0000095951 | |
| Sequenced-based reagent | *Piezo1* | Life Technologies | Mm01241549_m1 | |
| Sequenced-based reagent | *Piezo2* | Life Technologies | Mm01265861_m1 | |
| Sequenced-based reagent | *Ptgs2* | Life Technologies | Mm00478374_m1 | |

*Continued on next page*

Continued

| Reagent type (species) or resource | Designation | Source or reference | Identifiers | Additional information |
|---|---|---|---|---|
| Sequenced-based reagent | *Cyr61* | Life Technologies | Mm00487498_m1 | |
| Sequenced-based reagent | *Wnt1* | Life Technologies | Mm01300555_g1 | |
| Sequenced-based reagent | *Yap1* | Life Technologies | Mm01143263_m1 | |
| Sequence-based reagent | *Taz* | Life Technologies | Mm01289583_m1 | |
| Sequence-based reagent | *Tnfsf11* | Life Technologies | Mm00441906_m1 | |
| Sequence-based reagent | *Tnfrsf11b* | Life Technologies | Mm00435452_m1 | |
| Sequence-based reagent | *Sost* | Life Technologies | Mm00470479_m1 | |
| Sequence-based reagent | *Mrps2* | Life Technologies | Mm00475529_m1 | |
| Sequence-based reagent | *Piezo1* sgRNA | This paper | | GGTTATTCCTGTGAGGCCCG |
| Sequence-based reagent | *Piezo1* sgRNA | This paper | | TTAGGATTCGGCTCACAGAG |
| Chemical compound, drug | Yoda1 | Sigma-Aldrich | Cat# SML1558 | |
| Chemical compound, drug | Puromycin dihydrochloride | Sigma-Aldrich | Cat# P8833 | |
| Chemical compound, drug | G418 disulfate | Sigma-Aldrich | Cat# G8168 | |
| Antibody | YAP1 | Cell Signaling | Cat# 14074S; RRID:AB_2650491 | 1:200 |
| Antibody | Goat anti-Rabbit IgG (Alexa Fluor 488) | Abcam | Cat# ab150077; RRID:AB_2630356 | 1:200 |
| Software, algorithm | Prism 8 | GraphPad | https://www.graphpad.com/scientific-software/prism/ | |
| Software, algorithm | ImageJ | NIH | https://imagej.nih.gov/ij | |

## Mice

The generation of mice harboring *Piezo1* conditional allele, termed *Piezo1^{f/f}* mice, was described previously (*Cahalan et al., 2015*). Mice harboring both *Yap1* and *Taz* conditional alleles, termed *Yap1^{f/f};Taz^{f/f}* mice were kindly provided by Eric N. Olson (UT Southwestern Medical Center, Texas) and were described previously (*Xin et al., 2013*). The 8 kb Dmp1-Cre transgenic mice were described previously (*Bivi et al., 2012*). To generate *Dmp1-Cre; Piezo1^{f/f}* mice and littermates, we mated *Piezo1^{f/f}* mice (crossed into C57BL/6J for more than 10 generations) and *Dmp1-Cre* mice (crossed into C57BL/6J for more than 10 generations). *Dmp1-Cre; Yap1^{f/f},Taz^{f/f}* mice and littermates were obtained by mating *Yap1^{f/f},Taz^{f/f}* mice (mixture of 129/Sv and C57BL/6J) and *Dmp1-Cre* mice (crossed into C57BL/6J for more than 10 generations). We housed all mice in the animal facility of the University of Arkansas for Medical Sciences. Animal studies were performed in strict accordance with the recommendations in the Guide for the Care and Use of Laboratory Animals of the National Institutes of Health. Animal use protocols (3782, 3805, and 3897) were approved by the Institutional Animal Care and Use Committee (IACUC) of the University of Arkansas for Medical Sciences. All of the animals were handled according to approved protocols.

To quantify cancellous bone formation, we injected mice with calcein (20 mg/kg body weight) intraperitoneally 7 and 3 days before harvest. To quantify periosteal and endocortical bone formation, we injected mice with calcein (20 mg/kg body weight) and Alizarin Red (20 mg/kg body weight)

10 and 3 days before harvest. For gene expression, we injected Yoda1 into 4-month-old female C57BL/6J mice one time and harvested tibiae 4 hr later for RNA extraction. For bone mass evaluation, we injected Yoda1 into 4-month-old female C57BL/6J mice five consecutive days per week for 2 weeks (day 1–5 and day 8–12) and harvested the mice at day 15 for analysis. Yoda1 (Sigma, St. Louis, MO) was dissolved in DMSO at 40 mM as a stock, diluted in 5% ethanol, and injected intraperitoneally at 5 µmol/kg body weight. Mice were rank-ordered by body weight and then assigned to Veh or Yoda1 groups to give identical group means. All investigators involved in data collection were blinded as to the genotype and group of the mice.

## Cell line

HEK 293 T cells were authenticated by ATCC. MLO-Y4 cells were created and authenticated in Dr. Lynda Bonewald's lab (*Kato et al., 1997*). We tested the MLO-Y4 cells by morphology and osteocytic gene expression such as RANKL and OPG. UAMS-32 cells were created and authenticated by Dr. Charles O'Brien (*O'Brien et al., 1999*; *Fu et al., 2002*). Cells were treated with plasmocin to prevent potential mycoplasma contamination.

## Cell cultures

MLO-Y4 cells were cultured in α-MEM supplemented with 5% FBS, 5% BCS, and 1% penicillin/streptomycin/glutamine. Fifteen dynes/cm$^2$ oscillatory fluid shear stress was applied on MLO-Y4 cells at 1 Hz for 2 hr using an IBDI pump system (IBIDI, Germany). For Yoda1 treatment, cells were cultured in the presence of 10 µM Yoda1 (Sigma, St. Louis, MO) or DMSO for 2 hr. Immediately after the treatments, we isolated RNA from cells using RNeasy mini kit (Qiagen, Germany) for qPCR or RNA-seq analysis. To silence *Piezo1*, we generated *Piezo1* shRNA expression plasmid using the following oligonucleotides in the pLKO.1-TRC cloning vector (Addgene Plasmid #10878, a gift from David Root): forward oligo: 5'-CCGGTC-GGCGCTTGCTAGAACTTCACTCGAGTGAAGTTCTAGCAAGCGCCGA TTTTTG-3'; reverse oligo: 5'- AATTCAAAAATCGGCGCTTGCTAGAACTTCACTCGAGTGAAGTTC TAGCAAGCGC-CGA-3' (*Zhang et al., 2017*). *Yap1* shRNA (TRCN0000238432) and *Taz* shRNA (TRCN0000095951) were purchased from Sigma (St. Louis, MO). A shRNA against firefly luciferase was used as a control (Sigma, St. Louis, MO). For virus production, HEK293T cells were cultured in a 6-well culture plate and co-transfected with a total 3 µg of lentiviral shRNA vector, pMD2G (Addgene plasmid #12259, a gift from Didier Trono), and psPAX2 (Addgene plasmid # 12260, a gift from Didier Trono) at the ratio of 2:0.9:0.4 using TransIT-LT1 transfection reagent (Mirus, Madison, WI). Culture media was changed 12 hr after transfection and viral supernatants were collected 48 hr after media change. Viral supernatants were filtered through a 0.45 µm filter and used immediately to transduce cells cultured in a 10 cm dish. Cells were then subjected to selection with G418 (100 µg/ml) or puromycin (25 µg/ml) for 5 days before treatment. To overexpress Piezo1 in MLO-Y4 cells, we transfected mPiezo1-IRES-eGFP (Addgene plasmid # 80925, a gift from Ardem Patapoutian) into MLO-Y4 cells using TransIT-LT1 transfection reagent (Mirus, Madison, WI) and then treated these cells with 15 dynes/cm$^2$ oscillatory fluid shear stress at 1 Hz for 2 hr. Plasmids for expression of *Cas9* and sgRNAs for knocking out *Piezo1* in UAMS-32 cells were prepared by inserting oligonucleotides encoding the desired sgRNA sequence into the pX458 vector using the protocol recommended by the Zhang laboratory (*Cong et al., 2013*). Plasmids expressing *Cas9* and *Piezo1* sgRNAs were transfected into UAMS-32 cells using TransIT-LT1 transfection reagent (Mirus, Madison, WI). Cells were sorted into 96-well plates for single cell cloning 48 hr after transfection. Single cell colonies were then screened for homozygous deletion using the following primers: Forward: 5'-GCTGTCAGGG TAAGCAGTATC-3', Reverse: 5'-GGAATATGAGGACAGCAGTCC-3'. All homozygous mutant cell colonies were then pooled together for further analysis. *Cas9* transfected cells were used as a control. All in vitro cell culture experiments were performed three times with three technical replicates.

## Femoral organ culture

Female mice at 5 weeks of age were euthanized in a CO$_2$ chamber. Femurs were dissected and both ends were removed in a culture hood. Bone marrow was then flushed out using PBS and the periosteal surface was scraped to remove periosteal cells. Femoral shafts were then cultured in a 12-well-plate with 1 ml of α-MEM supplemented with 10% FBS and 1% penicillin/streptomycin/glutamine for 24 hr. We then treated femur shafts with 10 µM Yoda1 (Sigma, St. Louis, MO) or DMSO for 4 hr.

Femur shafts were then collected for RNA isolation and qPCR analysis. Ex vivo femoral organ culture was repeated twice with three biological replicates.

## In vitro osteoblast differentiation

Bone marrow stromal cells were flushed out from long bones, collected into a 50 ml cubical tube, and filtered through a 40 μm cell strainer to obtain a single cell suspension. Bone marrow stromal cells were then seeded into a 12-well-plate at $5 \times 10^6$ cells/well and cultured in α-MEM containing 10% fetal bovine serum, 1% penicillin/streptomycin/glutamine, 1% ascorbic acid, and 10 mM β-glycerolphosphate. Culture medium was changed every 3 days. After 21 days, the cultures were fixed with 10% buffered formalin and stained with an aqueous solution of 40 mM Alizarin Red to evaluate osteoblastogenesis.

## RNA-seq analysis

Purified RNA was used as input for sequencing library preparation and indexing using the TruSeq stranded mRNA kit (Illumina, CA), following the manufacturer's protocol. The libraries were then pooled and sequenced using a NextSeq sequencer with 75 cycles of sequencing reaction. Data handling and processing were performed on the basis of a previous bioinformatics pipeline (*Nookaew et al., 2012*). The high-quality reads (phred quality score,>25; length after trimming,>20 bases) were obtained with the dynamic trimming algorithm in the SolexaQA++ toolkits (*Cox et al., 2010*), and aligned with the mouse genome version GRCm38 using BWA software (*Li and Durbin, 2009*). Then the alignment files (.bam) were used to generate read counts for statistical analysis. The differential gene expression analysis was performed using negative binomial based statistic (*Love et al., 2014*). The adjusted p-values were used for gene enrichment analysis based on Gene Ontology using the piano package (*Väremo et al., 2013*). Raw RNA-seq results have been deposited in GEO database under BioProject PRJNA551282 with accession numbers: SRR9598498, SRR9598497, SRR9598496, SRR9598495, SRR9598494, and SRR9598493. Detailed RNAseq analysis was shown in *Supplementary file 1*.

## Calcium concentration measurement

For intracellular calcium concentration measurement under fluid flow condition, $1 \times 10^5$ MLO-Y4 cells were seeded in a μ-Slide I Luer (0.4 mm) fluid chamber slide (IBIDI, Germany) overnight. One hour before initiating fluid flow, the culture medium was removed and 100 μl Hank's Buffer with Hepes (HHBS) containing 4 μM Fluo-8 (Abcam, Cambridge, MA) was added to the culture, as described by the manufacturer. The cells were then cultured at 37℃ for 30 min. After additional incubation at room temperature for 30 min, the chamber slide was placed under a confocal microscope in order to record the intensity of fluorescence of MLO-Y4 cells. Fluorescence was recorded for 3 min before starting fluid flow using HHBS and then recorded for another 10 min. The increase of intracellular concentration was calculated by subtracting the initial mean fluorescence. For measuring intracellular calcium concentration in cells with Yoda1 treatment, we cultured 4,000 MLO-Y4 cells per well in a 96-well-plate. We preloaded the cells with Fluo-8 as described by the manufacturer and read the intensity of the fluorescence using a Victor X3 multi-label plate reader (Perkin Elmer, Waltham, MA) immediately after the treatment. We measured the fluorescence for 5 min at an interval of 20 s. The percentage of increase in intracellular calcium concentration was calculated as $(F_x-F_0)/F_0$.

## Skeletal analysis

Tibial X-rays were obtained using an UltraFocus X-ray machine (Faxitron Bioptics, Tucson, Arizona) and BMD of the lumbar spine and femur were measured by dual-energy X-ray absorptiometry using a PIXImus Densitometer (GE-Lunar Corp.) Three dimensional bone volume and architecture of L4 vertebra, femur, and tibia were measured by μCT (model μCT40, Scanco Medical, Wayne, PA). The femur, vertebrae (L4), or tibia, were cleaned of soft tissues and fixed in 10% Millonig's formalin for 24 hr. Bone were then gradually dehydrated into 100% ethanol. Bone samples were loaded into a 12.3 mm diameter scanning tube and images acquired in the μCT40. The scans were integrated into 3D voxel images (1024 × 1024 pixel matrices for each individual planar stack) and a Gaussian filter (sigma = 0.8, support = 1) was used to reduce signal noise. Scanco Eval Program v.6.0 was used for

measuring bone volume. A threshold of 220 mg/cm$^3$ was applied to all scans at medium resolution (E = 55 kVp, I = 145 μA, integration time = 200 ms) for trabecular bone measurements. The cortical bone and the primary spongiosa were manually excluded from the analysis. Trabecular bone measurements at the distal femur were made on 151 slices beginning 8–10 slices away from the growth plate and proceeding proximally. Trabecular bone measurements in the vertebra was determined using 100 slices (1.2 mm) of the anterior (ventral) vertebral body immediately inferior (caudal) to the superior (cranial) growth plate. All trabecular measurements were made by drawing contours every 10 to 20 slices and voxel counting was used for bone volume per tissue volume and sphere filling distance transformation indices, without pre-assumptions about the bone shape as a rod or plate for trabecular microarchitecture. Femoral cortical thickness, periosteal circumference, and endocortical circumference were measured at the mid-diaphysis. For tibial cortical thickness, we analyzed 18 slices 5 mm proximal from the distal tibiofibular junction. Vertebral cortical bone thickness was determined on the ventral cortical wall using contours of cross-sectional images, drawn to exclude trabecular bone. Cortical analysis were measured at a threshold of 260 mg/cm$^3$. Calibration and quality control were performed weekly using five density standards and spatial resolution was verified monthly using a tungsten wire rod. We based beam-hardening correction on the calibration records. Corrections for 200 mg hydroxyapatite were made for all energies.

## Histology

Lumbar vertebrae were fixed for 24 hr in 10% Millonig's formalin, dehydrated into 100% ethanol, embedded in methyl methacrylate, and then 5 μm longitudinal sections were obtained. After removal of plastic and rehydration, we stained sections for TRAP activity and counter-stained with T-blue. Quantitative histomorphometry was performed to determine osteoblast and osteoclast number using Osteomeasure system (OsteoMetrics, Decatur, GA) interfaced to an Axio image M2 (Carl Zeiss, NY). Bone formation rate was measured using unstained sections in Osteomeasure system. We used terminology recommended by the Histomorphometry Nomenclature Committee of the American Society for Bone and Mineral Research (*Dempster et al., 2013*). For quantification of periosteal and endocortical bone formation, femurs or tibiae were fixed in 10% Millonig's formalin for 24 hr, dehydrated into 100% ethanol, embedded in methyl methacrylate, and then 80 μm cross sections were obtained at the femoral mid-diaphysis for femoral sections and 5 mm proximal from the distal tibiofibular junction for tibial sections. We then measured mineralizing surface and mineral apposition rate using the Osteomeasure system.

## Tibia axial loading

A cyclic axial load was applied to left tibia of mice to achieve +1200 με peak strain at the tibial midshaft using an Electroforce TA 5500 (TA Instruments, New Castle, DE). To determine the required load to achieve +1200 με peak strain for each genotype of experimental mice, axial loading was applied to harvested tibiae ex vivo. A single-element strain gauge (C2A-06-015LW-120, VPG Micro-Measurements, Wendell, NC) was attached to the antero-medial surface of the tibia located 5 mm proximal from the distal tibiofibular junction using M-Bond 200 adhesive kit (VPG Micro-Measurements). We recorded the force-strain regressions using Electroforce TA 5500 software. We then applied the same amount of load to mice in vivo according to their genotype (8.5 Newton for *Piezo1$^{f/f}$* mice and 7.5 Newton for *Dmp1-Cre; Piezo1$^{f/f}$* mice). The left tibia of each mouse was loaded for five consecutive days per week for 2 weeks (day 1–5 and day 8–12), and the load was applied in 1200 cycles with 4 Hz triangle waveform and 0.1 s rest time between each cycle, a protocol shown to be anabolic (*Sun et al., 2018*). We injected calcein (Sigma, St. Louis, MO) and Alizarin Red (Sigma) intraperitoneally into mice 10 days and 3 days before euthanasia to label new bone formation. We euthanized the mice and collected tissues at day 15 for skeletal analysis. For gene expression analysis, we loaded left tibia of 4-month-old female mice with one bout of loading and harvested tibiae 5 hr after loading for RNA extraction.

## Biomechanical testing

We performed three-point bending test on femurs at room temperature using a miniature bending apparatus with the posterior femoral surface lying on lower supports (7 mm apart) and the left support immediately proximal to the distal condyles. Load was applied to the anterior femoral surface

by an actuator midway between the two supports moving at a constant rate of 3 mm/min to produce a physiological in vivo strain rate of 1% for the average murine femur. Maximum load (N) and displacement (mm) were recorded. The external measurements (length, width and thickness) of the femora were recorded with a digital caliper. We measured the moment of inertia in the midshaft of femur using μCT (model μCT40, Scanco Medical). The mechanical properties were normalized for bone size and ultimate strength and stress (N/mm$^2$; in megapascals and MPa) was calculated.

## Quantitative PCR

Organs and whole bones were harvested from animals, removed of soft tissues, and stored immediately in liquid nitrogen. We prepared osteocyte-enriched bone by removing the ends of femurs and tibias and then flushing the bone marrow with PBS. We then scraped the bone surface with a scalpel and froze them in liquid nitrogen for later RNA isolation, or decalcified them for genomic DNA isolation. We isolated total RNA using TRIzol (Life Technologies, NY), according to the manufacturer's instructions and prepared cDNA using High Capacity first strand cDNA synthesis kit (Life Technologies). We performed quantitative RT-PCR using the following Taqman assays from Applied Biosystems: *Piezo1* (Mm01241549_m1); *Piezo2* (Mm01265861_m1); *Ptgs2* (Mm00478374_m1); *Cyr61* (Mm00487498_m1); *Wnt1* (Mm01300-555_g1); *Yap1* (Mm011432-63_m1); *Taz* (Mm01289583_m1); *Tnfsf11* (Mm00441906_m1); *Tnfrsf11b* (Mm00435452_m1); *Sost* (Mm00470479_m1); and ribosomal protein S2 (*Mrps2*) (Mm00475529_m1). We calculated relative mRNA amounts using the ΔCt method (*Livak and Schmittgen, 2001*). We isolated genomic DNA from decalcified cortical bone after digestion with proteinase K and phenol/chloroform extraction. We obtained two custom Taqman assays from Applied Biosystems for quantifying *Piezo1* gene deletion efficiency: one specific for sequences between the loxP sites and the other specific for sequences downstream from the 3′ loxP site.

## Immunostaining

Cultured cells were fixed in 4% freshly prepared paraformaldehyde for 15 min. Slides were washed in PBST for 5 min, pretreated with PBS containing 0.1% Triton X-100 for 20 min, and blocked in 2.5% normal goat serum for one hour. Anti-YAP1 antibody (14074S, Cell Signaling, Danvers, MA) was diluted 1:200 in PBST containing 2.5% normal goat serum and incubated with the chamber slides overnight at 4°C followed by rinsing and additional incubation for 1 hr with goat anti-rabbit IgG H and L (Alexa Fluor 488) (1:200) (ab150077, Abcam, Cambridge, MA). Non-immune goat IgG was used as a negative control. Slides were mounted with aqueous mounting medium (H-1000, VECTOR LABORATORIES, INC., Burlingame, CA). Stained slides were imaged using Axio imager M2 fluorescence microscope (Carl Zeiss, NY). Mean fluorescence intensity was quantified using ImageJ (NIH, Bethesda, Maryland).

## Osteocalcin and CTX ELISA

Circulating osteocalcin and CTX in serum was measured using a mouse Osteocalcin enzyme immunoassay kit (Thermo Fisher) and RatLaps (CTX-I) EIA kit (Immunodiagnostic Systems, Boldon, UK) respectively according to the manual provided by manufacturers. Blood was collected by retro-orbital bleeding into 1.7 mL microcentrifuge tubes. Blood was then kept at room temperature for one hour and centrifuged at 1500 x g for 10 min to separate serum from cells.

## Statistical analysis

GraphPad Prism seven software (GraphPad, San Diego) was used for statistical analysis. Two-way analysis of variance (ANOVA) or Student's t-test were used to detect statistically significant treatment effects, after determining that the data were normally distributed and exhibited equivalent variances. All t-tests were two-sided. *P*-values less than 0.05 were considered as significant. Error bars in all figures represent s.d..

## Acknowledgements

We thank CA O'Brien for valuable discussions and advice; IB Gubrij, J Kordsmeier, JA Crawford, RD Peek, SB Berryhill, WR Hogue and JJ Goellner for technical support; AG Robling (Indiana University School of Medicine) for the tibia cross sectioning protocol, the CTPR Developmental Genomics

Core, the UAMS Digital Microscopy Core, and the staff of the UAMS Department of Laboratory Animal Medicine. This work was supported by the National Institute of General Medical Sciences (NIGMS) grants P20GM125503 and P20GM121293; the National Institute of Arthritis and Musculoskeletal and Skin Diseases (NIAMS) grants R01AR56679 and R01AR047867; and the UAMS Bone and Joint Initiative.

## Additional information

### Funding

| Funder | Grant reference number | Author |
|---|---|---|
| National Institute of General Medical Sciences | P20GM125503 | Intawat Nookaew<br>Erin Mannen<br>Maria Almeida<br>Jinhu Xiong |
| National Institute of Arthritis and Musculoskeletal and Skin Diseases | R01AR56679 | Maria Almeida |
| National Institute of Arthritis and Musculoskeletal and Skin Diseases | R01AR047867 | Matthew J Silva |
| University of Arkansas for Medical Sciences | Bone and Joint Initiative | Jinhu Xiong |

The funders had no role in study design, data collection and interpretation, or the decision to submit the work for publication.

### Author contributions

Xuehua Li, Data curation, Formal analysis, Validation, Investigation, Methodology; Li Han, Data curation, Formal analysis, Investigation; Intawat Nookaew, Data curation, Formal analysis; Erin Mannen, Data curation, Formal analysis, Methodology; Matthew J Silva, Methodology; Maria Almeida, Resources, Writing—review and editing; Jinhu Xiong, Conceptualization, Formal analysis, Supervision, Investigation, Methodology, Writing—original draft, Writing—review and editing

### Author ORCIDs

Erin Mannen (iD) http://orcid.org/0000-0003-4441-5411
Jinhu Xiong (iD) https://orcid.org/0000-0002-7270-9284

### Ethics

Animal experimentation: Animal studies were performed in strict accordance with the recommendations in the Guide for the Care and Use of Laboratory Animals of the National Institutes of Health. Animal use protocols (3782, 3805, and 3897) were approved by the Institutional Animal Care and Use Committee (IACUC ) of the University of Arkansas for Medical Sciences. All of the animals were handled according to approved protocols.

### Decision letter and Author response

Decision letter https://doi.org/10.7554/eLife.49631.021
Author response https://doi.org/10.7554/eLife.49631.022

## Additional files

### Supplementary files

• Supplementary file 1. RNAseq analysis of MLO-Y4 cells cultured under fluid shear stress (FF) and static (ST) conditions.
DOI: https://doi.org/10.7554/eLife.49631.016

• Transparent reporting form
DOI: https://doi.org/10.7554/eLife.49631.017

## Data availability

RNAseq source data for Figure 1A, Figure 1—figure supplements 1 and 2, and Supplementary file 1 has been deposited in BioProject under accession code PRJNA551282.

The following dataset was generated:

| Author(s) | Year | Dataset title | Dataset URL | Database and Identifier |
|---|---|---|---|---|
| Xuehua Li, Han Li, Intawat Nookaew, Erin Mannen, Matthew J. Silva, Maria Almeida, Jinhu Xiong | 2019 | Influence mechanical signals promotes bone anabolism via Piezo1 | http://www.ncbi.nlm.nih.gov/bioproject/?term=PRJNA551282 | NCBI Bioproject, PRJNA551282 |

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
