## [Decision Letter]

Thank you for submitting your article "Stimulation of Piezo1 by mechanical signals promotes bone anabolism" for consideration by *eLife*. Your article has been reviewed by two peer reviewers, and the evaluation has been overseen by Anna Akhmanova as the Senior and Reviewing Editor. The following individual involved in review of your submission has agreed to reveal their identity: Nele A Haelterman (Reviewer #1).

The reviewers have discussed the reviews with one another and the Reviewing Editor has drafted this decision to help you prepare a revised submission.

This is a well-written manuscript describing a role for the mechano-sensitive ion channel Piezo1 in bone remodeling. The authors find that conditional deletion of Piezo1 in the osteoblast/osteocyte lineage decreases cortical thickness as well as trabecular bone volume. In addition, loss of this channel abolishes mechanical-induced bone accrual, suggesting that Piezo1 is a key factor in regulating this aspect of bone homeostasis. Finally, the authors show that Piezo1 functions through Yap/Taz, which activate Wnt1 signaling to increase bone mass. The data presented in this manuscript are compelling. Very recently, a paper by Sun et al. has similarly identified Piezo1 as a major player in bone mechanosensation. Both groups have independently come to the same conclusion, which strengthens the finding that Piezo1 plays a role in this process. The current manuscript still provides a significant advance to the field, given that the authors provide mechanistic insights into the pathways that are activated upon mechanical loading of bone. In addition, the current manuscript also answers whether Piezo1 is truly required for bone's mechanosensing ability by performing anabolic loading-experiments on conditional Piezo1 mutants.

Essential revisions:

1) The authors suggest a model in which mechanical stimulation activates Yap/Taz transcription factors to induce Wnt1 expression and hereby increase bone mass. Indeed, in conditional Piezo1 mutants Wnt1-expression is reduced. To unequivocally prove that in vivo, this pathway is indeed underlying the anabolic response to mechanical stimuli, the authors should return to the samples collected from their anabolic bone-loading experiment and: a) Show that Wnt-1 expression is increased in Piezo1f/f mice upon loading, whereas this increase is blunted in Dmp1-Cre; Piezo1f/f mice. This could be done by Q-RT PCR.

b) Show a change in the nuclear localization of Yap/Taz in Piezo1f/f mice upon loading, that is missing from Dmp1-Cre; Piezo1f/f mice.

2) The authors perform RNA-seq to determine the effects of FSS on global patterns of gene expression in MLO-Y4 cells, but then only look at a few select genes (Wnt1, Ptgs2, Opg) for most of the subsequent in vitro experiments after Piezo1 or Yap/Taz knockdown +/- Yoda1 treatment. Ideally, the authors would perform additional transcriptomic profiling in these studies to determine the overall impact of Piezo1 deficiency on global patterns of gene expression changes in response to FSS. In the absence of these data, extending their panel of RT-qPCR target genes would be helpful to better understand how important Piezo1 is for gene expression changes in response to FSS signaling in vitro.

3) In Figure 3C, D, it is very interesting that Piezo1 conditional knockout mice show clear (only slightly blunted) increases in loading-induced bone formation, yet no detectable gains in bone mass. This apparent discrepancy should be explored in more detail. What are the effects of loading on osteoclastic bone resorption in these animals? It would be sufficient to show serum CTX levels.

4) The in vitro data obtained with Yoda1 in Figure 5E-H are interesting, but appear incomplete. Overall, the effects of this treatment on bone mass are rather mild, such that a large (n=12/group) sample size was needed to observe statistically-significant effects. Given potential effects of Piezo1 on osteoclasts, it is important to measure serum CTX levels. Finally, an opportunity appears to have been missed here to determine the effects of Yoda1 on bone endpoints in Piezo1 conditional knockout mice. This would be a nice, though not an essential experiment to link the in vitro and in vivo data, and to address the specificity of Yoda1 in mice.

5) The impact of Piezo1 deficiency on Ptgs2 up-regulation in Figure 1E is quite mild. This strongly suggests that other (Piezo1-independent) pathways are needed for Ptgs2 upregulation. The authors should at least expand their discussion to consider how other signaling pathways may be contributing to gene expression changes in response to FSS.

---

## [Author Response]

Essential revisions:1) The authors suggest a model in which mechanical stimulation activates Yap/Taz transcription factors to induce Wnt1 expression and hereby increase bone mass. Indeed, in conditional Piezo1 mutants Wnt1-expression is reduced. To unequivocally prove that in vivo, this pathway is indeed underlying the anabolic response to mechanical stimuli, the authors should return to the samples collected from their anabolic bone-loading experiment and: a) Show that Wnt-1 expression is increased in Piezo1f/f mice upon loading, whereas this increase is blunted in Dmp1-Cre; Piezo1f/f mice. This could be done by Q-RT PCR.

For the tibia anabolic loading experiment, we loaded the left tibia and used the right tibia as a control. The tibiae from this experiment have been sectioned and used for histomorphometry analysis to measure bone formation rate. Thus, there were not samples available to perform gene expression analysis. To address the reviewer’s question 1a), we have performed another short-term tibia compressive loading experiment and measured *Wnt1* expression in the tibia of these mice. We found that mechanical loading increased *Wnt1* expression in control mice. Importantly, *Piezo1* deletion in Dmp1-Cre targeted cells blunted the increase of *Wnt1* induced by mechanical loading. We have included these data in Figure 4G in the revised manuscript.

b) Show a change in the nuclear localization of Yap/Taz in Piezo1f/f mice upon loading, that is missing from Dmp1-Cre; Piezo1f/f mice.

We have made several attempts at immunostaining of YAP1/TAZ in bone sections and have not obtained good signal to noise ratio. As an alternative, we measured *Cyr61*, a YAP1/TAZ target gene, expression in mice to evaluate YAP1/TAZ activity in vivo. We found that *Cyr61* expression in cortical bone was upregulated by mechanical loading in control mice and this increase was blunted in *Dmp1-Cre; Piezo1^f/f^* mice. This result is consistent with the idea that YAP1/TAZ activity was increased by mechanical stimulation and that loss of Piezo1 in Dmp1-Cre targeted cells blunted this increase in YAP1/TAZ activity, supporting the idea that Piezo1 controls YAP1/TAZ activity by responding to mechanical stimulation. We have included these data in the Figure 4G in the revised manuscript.

2) The authors perform RNA-seq to determine the effects of FSS on global patterns of gene expression in MLO-Y4 cells, but then only look at a few select genes (Wnt1, Ptgs2, Opg) for most of the subsequent in vitro experiments after Piezo1 or Yap/Taz knockdown +/- Yoda1 treatment. Ideally, the authors would perform additional transcriptomic profiling in these studies to determine the overall impact of Piezo1 deficiency on global patterns of gene expression changes in response to FSS. In the absence of these data, extending their panel of RT-qPCR target genes would be helpful to better understand how important Piezo1 is for gene expression changes in response to FSS signaling in vitro.

We have measured a set of genes that are known to control osteoblast formation in tibial cortical bone of *Dmp1-Cre;Piezo1^f/f^* and *Piezo1^f/f^* mice, including *Sost, Dkk1, Wnt1, Wnt7b, Wnt16, Bmp2*, and *Tgfb1*. We only observed a change in *Wnt1* expression (Author response image 1). We measured the same set of genes after *Piezo1* knockdown in MLO-Y4 cells cultured under static or fluid flow conditions. Only *Wnt1, Dkk1* and *Tgfb1* were expressed in MLO-Y4 cells at a measurable level. The other genes were not detectible in MLO-Y4 cells. Consistent with our in vivo results, only *Wnt1* expression was affected by *Piezo1* deletion. We agree with the reviewers that an unbiased transcriptomic analysis of cells lacking Piezo1 in response to fluid shear stress will provide more information to better understand how Piezo1 mediates the response to mechanical stimulation. However, we would like to suggest that this would be appropriate as a future study and that lack of this information will not compromise the main conclusions in our current study.

**Author response image 1. respfig1:** Gene expression analysis of *Piezo1*-deficient cells. (**A**) RT-qPCR analysis of *Sost, Dkk1, Wnt1, Wnt7b, Bmp2*, and *Tgfb1* in tibia cortical bone of12-week-old female *Piezo1^f/f^* (n = 9) and *Dmp1-Cre;Piezo1^f/f^* (n = 9) mice. **p* < 0.05 using Student’s t-test. (**B**) *Wnt1, Dkk1*, and *Tgfb1* mRNA levels in control or *Piezo1* knock-down MLO-Y4 cells cultured under static or fluid shear stress conditions. **p* < 0.05 with the comparisons indicated by the brackets using 2-way ANOVA.

3) In Figure 3C, D, it is very interesting that Piezo1 conditional knockout mice show clear (only slightly blunted) increases in loading-induced bone formation, yet no detectable gains in bone mass. This apparent discrepancy should be explored in more detail. What are the effects of loading on osteoclastic bone resorption in these animals? It would be sufficient to show serum CTX levels.

In our tibia compressive loading experiment, only the left tibia was loaded. The changes occurring locally at the loaded tibia may not contribute significantly to circulating serum markers such as CTX. In addition, previous studies have shown that anabolic loading does not affect osteoclast number (Brodt and Silva, 2010) and osteoclast marker gene expression, such as Cathepsin K (Silva et al., 2012), indicating that anabolic loading as performed here does not affect bone resorption. The fact that there still was a significant increase in bone formation in loaded tibia in the conditional knockout mice indicates that Piezo1 is not the only mechanosensor in bone. Other pathways such as integrins or primary cilia may also contribute to the skeletal response to mechanical stimulation. Importantly, the bone formation induced by anabolic loading in *Piezo1* conditional knockout mice was not sufficient to cause a measurable increase in cortical thickness, such as occurred in control mice, demonstrating the importance of Piezo1 in mediating the skeletal response to mechanical loading.

4) The in vitro data obtained with Yoda1 in Figure 5E-H are interesting, but appear incomplete. Overall, the effects of this treatment on bone mass are rather mild, such that a large (n=12/group) sample size was needed to observe statistically-significant effects. Given potential effects of Piezo1 on osteoclasts, it is important to measure serum CTX levels. Finally, an opportunity appears to have been missed here to determine the effects of Yoda1 on bone endpoints in Piezo1 conditional knockout mice. This would be a nice, though not an essential experiment to link the in vitro and in vivo data, and to address the specificity of Yoda1 in mice.

We have measured the serum CTX level and found that it did not change in Yoda1 treated mice compared to vehicle treated mice. We have included these data in Figure 5—figure supplement 1. We agree that administration of Yoda1 to Piezo1 conditional knockout mice would nicely address the specificity of the compound for Piezo1. Unfortunately, production of adult mice for this experiment, drug administration, and skeletal analysis would require significantly more time than the 2 months suggested for submission of revisions.

5) The impact of Piezo1 deficiency on Ptgs2 up-regulation in Figure 1E is quite mild. This strongly suggests that other (Piezo1-independent) pathways are needed for Ptgs2 upregulation. The authors should at least expand their discussion to consider how other signaling pathways may be contributing to gene expression changes in response to FSS.

We agree with the reviewers that Piezo1 is not the sole mechanosensor that mediates the response to mechanical stimulation in bone cells. In addition to Piezo1, other cell surface proteins and structures, including integrins, focal adhesions, and primary cilia, may also contribute to sensing mechanical signals in bone. The crosstalk between Piezo1 and these other sensors needs to be addressed in future studies. We have discussed this issue in the revised manuscript.